# A Characteristic Analysis of Various Air Pollutants and Their Correlation with O₃ in the Jiangsu, Shandong, Henan, and Anhui Provinces of China

**Tianzhen Ju** [1],*, **Bingyu Pan** [1], **Bingnan Li** [2], **Jiwei Wang** [1], **Shuya Liu** [1], **Shuai Peng** [1] and **Meng Li** [1]

[1] College of Geography and Environmental Sciences, Northwest Normal University, Lanzhou 730070, China
[2] Faculty of Atmospheric Remote Sensing, Shaanxi Normal University, Xi'an 710062, China
* Correspondence: jujutz@163.com; Tel.: +86-15811080907

**Abstract:** Using a series of characteristic analyses of HCHO, SO₂, NO₂, O₃, and absorptive aerosols (ultraviolet aerosol index—UVAI) in the atmosphere of the study area from 2011 to 2021, this paper reviews the pollutants present in Jiangsu, Shandong, Henan, and Anhui provinces, which are the key regions for air pollution control in China. Furthermore, the correlation between various air pollutants and O₃ is also examined. The results obtained show the following: (1) Within the research areas, temporal pollutant variations can be observed from 2011 to 2021. NO₂ and HCHO only modestly decrease, SO₂ sharply declines, while UVAI remains stable. During the study period, HCHO and O₃ first increase and subsequently decline, whereas the other three pollutants exhibit the opposite behavior. (2) Among the examined pollutants, SO₂ is the most unstable. (3) Our research discovered that the atmospheric transport paths at high-value points significantly vary between seasons using the public backward trajectory model. (4) Through the correlation analysis we performed, this research reveals the different degrees of spatial correlations between O₃ and other pollutants. (5) Using the FNR index method, the sensitivity of O₃ to its precursors (NOx and VOCs) is investigated. Based on the results, we provide some suggestions concerning the primary control pollutants and relative control strategies in different seasons and regions.

**Keywords:** OMI; spatiotemporal distribution; backward trajectory; O₃ sensitivity

## 1. Introduction

In the context of global climate change, a variety of air pollutants are harmful to human health, worsen air quality, and have an impact on ecosystems; therefore, they have piqued the interest of scientists all over the world [1,2]. This includes extensive research on China, which, for a long period of time, has been playing the role of a producer in the global market and the production base of many developed countries, which has led to rapid economic development in China but has also caused serious environmental damage. In China, single-source air pollution, such as coal smoke and petrochemical pollution, has gradually evolved into complex air pollution. Traditional pollutants SO₂ and TSP have been well controlled, but the rapid increase in the number of motor vehicles has caused NOₓ emissions to continue to rise, and regional combined air pollution characterized by absorbent aerosols (for example, carbon aerosols, dust aerosols, etc.) and O₃ is becoming increasingly severe. When these pollutants accumulate in the air and exceed the natural degradation capacity, accumulation, diffusion, transfer, and interaction occur. When they interact, they produce many other secondary pollutants, which often cause various diseases of the human body, thus endangering the survival and development of human beings. Therefore, it is very important to master the dynamics of many kinds of air pollutants in a large area. The observation methods used for air pollutants mainly include satellite remote-sensing and ground-based observations. Ground monitoring features high-accuracy features and the simultaneous obtainment of multiple atmospheric environmental parameters; however,

ground-site monitoring is sparsely distributed and expensive to perform [3]. Such defects can be avoided in satellite remote-sensing observations [4,5]. Thanks to the free availability of satellite monitoring data in many countries, researchers can obtain data for analysis at a very low cost, and low-cost sensors can enable the high-density monitoring of air pollutants; the data obtained from large-scale monitoring over long time scales can be used to complement traditional pollution monitoring, improve exposure estimates, and raise community awareness of air pollution [6].

During the use of satellite monitoring data, many scholars have studied the change process of a single air pollutant. A study of atmospheric $NO_X$ in India conducted from 2007 to 2018 determined that Indian $NO_X$ gradually increased from 2007 to 2016, and only began to stabilize and slowly decrease in 2017. Additionally, $NO_X$ has a strong seasonal cycle, peaking in the summer [7]. Curier et al. [8] examined European $NO_X$ emission trends using OMI $NO_2$ tropospheric vertical column density data and the LOTOS-EUROS model. This evaluation provides a scientific evaluation algorithm. In the study of $SO_2$ in the Sichuan-Chongqing region of China, it was observed that the distribution of $SO_2$ in the region was also uneven, showing the characteristics of high pollution levels in the southeast and low pollution levels in the west, the overall fluctuation was minor, the stability was good, and the seasonal cycle was obvious: it was high in winter and low in summer. The distribution of $SO_2$ was closely related to PM2.5 [9]. Kang et al. [10] used satellite data to study the temporal and spatial distribution characteristics and variation trends of $SO_2$ in the global atmospheric boundary layer from 2005 to 2017. Duan et al. [11] investigated absorption aerosols in three northeastern provinces. The present study used the Hurst exponent analysis to explore and categorize the sustainability of future absorbing aerosols. In addition, the black carbon aerosols in the Chengdu region were studied, and the results show that different, seasonal air-quality sources produce different effects; however, this effect mainly occurs in the vicinity of Chengdu and the eastern region of Chongqing [12]. $O_3$ is a type of secondary pollutant that is primarily formed by the complex photochemical reaction of $NO_X$, CO, and VOCs in the presence of solar ultraviolet irradiation, and it is the main component of greenhouse gases and photochemical smog [13]. There are also many studies on monitoring $O_3$ using satellite data, for example, Yu Ruilian et al. summarized the formation mechanisms of atmospheric $O_3$, the spatiotemporal distribution characteristics of atmospheric $O_3$ in some areas of China, the relationship between atmospheric $O_3$ and its precursors, and the main factors affecting the concentration of atmospheric $O_3$ [14]. L Chen et al., using the total $O_3$ column data obtained from the European Center for Medium-Term Weather Forecast, analyzed the spatial and temporal distributions of the total $O_3$ column in the northwest Pacific by analyzing the spatial and temporal fields of the total $O_3$ column in the region, and the following conclusions were obtained. The low latitudes were the regions with the lowest total $O_3$ column, the zonal distribution was obvious, and the total $O_3$ column increased with the increase in latitude toward the Arctic [15]. Additionally, by using the reanalysis data of the ECMWF, the variation trend of $O_3$ in the Northern Hemisphere in the last 33 years was studied. Studies have shown that the total $O_3$ in four key regions in the Northern Hemisphere (Arctic, East Asia, and north and western Europe) showed a decreasing trend from 1979 to 2011, and that since 1993 and 1994, the total $O_3$ in winter in the four key regions in the Northern Hemisphere presents a significant upward trend, with the greatest upward trend visible in the Arctic region [16]. In the current paper, the progress of VOC monitoring is systematically introduced, and the main methods of VOC monitoring and evaluation are discussed from the aspects of monitoring and analysis methods, chemical-reaction activity, and health risk assessment. The characteristics of VOC concentration and source distribution in the areas where VOC research was conducted at home and abroad are compared and summarized [17].

It can be observed that many scholars have made use of satellite remote-sensing data to analyze different pollutants in different regions. Since the use of satellite data can analyze the changing process of atmospheric pollutants on larger and during longer time scales, although in a manner slightly less accurate than ground-based-site monitoring, the human,

material, and financial costs are significantly reduced, including in the study of large areas of air pollution, using satellite remote-sensing and ground station data. Many people have studied the process of change between single air pollutants and two or three air pollutants, but with the $O_3$ pollution problem becoming increasingly more serious since air pollution control no longer focuses on a single pollutant, it is necessary to study the relationship between $O_3$ and other common pollutants using long time-scale data.

The main point of the present paper is to study the spatial correlations between different pollutants and $O_3$ and to show whether this kind of pollutant is related to the change in and generation of $O_3$ through the correlation to assess the region in the multi-pollutant coordinated governance of key control indicators. Secondly, as an important air pollutant, HCHO is considered to be carcinogenic and genotoxic to living organisms and an important precursor of $O_3$ production in the environment; therefore, the human body as well as the ecological environment is exposed to non-negligible harm. In recent years, the research conducted on the concentration of HCHO in the atmosphere has decreased, and many scholars have focused on the detection and analysis of indoor HCHO [18]. Simultaneously, the temporal and spatial variations of atmospheric HCHO are analyzed by using remote-sensing data.

Finally, the research area presented in this paper is the newly designated key area of air pollution control in China, and it is also an important economic area in China, where both heavy and light industries are well developed. This area demonstrates obvious characteristics of regional air compound pollution, which limits the sustainable development of regional social economy in the future, causing harm to public health. However, previous studies have done little to analyze the region's atmospheric environment. As a result, the region must conduct an air pollution analysis [19,20].

We are inspired to work on this research primarily based on the government's attention to the study area and the necessity for environmentally sustainable development. This paper has clearly demonstrated which pollutants might be decreased by industrial transformation and other alternative activities, as well as in which regions it is required to restrict emissions of associated pollutants during a specific season to minimize air pollution, supplementing the studies on air pollution in this area and also offering some theoretical foundation for developing air pollution control measurements.

The paper is organized as follows: the "Research Area" section describes the research area's geographical location, natural conditions, economic development, etc.; the "Data and methods" section provides the research methods and formulas utilized in this study; the "Results and Discussion" section presents the findings of our study and discusses the complexity of the influencing factors of air pollution in the area and the association between diverse pollutants. Conclusions are given in the "Conclusions" section.

## 2. Research Area

The present study was conducted from 2011 to 2020, and the sample period was mainly selected because of China's five-year plans: 2011–2015 is the "12th Five-Year Plan" and 2016–2020 is the "13th Five-Year Plan"; different five-year plans have different policies, so there may be a turning point in atmospheric pollutant concentrations. With the introduction of China's 14th five-year plan, this research area (Figure 1) has also become another key area for air pollution control, following the Hebei and Fenwei Plains in Beijing and Tianjin. As an important economic zone in eastern China, there is a high research value for the distribution and transport of air pollutants. They are composed of the Jiangsu, Shandong, Henan, and Anhui provinces, and are located in eastern China. The study area's climate is classified as temperate continental monsoon. The northern section is the Beijing–Tianshui–Hebei economic circle, with the Chinese capital Beijing at its center. The eastern region has a long coastline, while the south is composed of the Yangtze River Delta economic zone of the Chinese economy, which is responsible for the important task of the Chinese economy's two-way opening. The cities in the study area are fairly densely packed. According to China's seventh population census, the total population of the four provinces was 347.2 million

at the end of 2020, accounting for 24.57% of China's total population. The GDP of the four provinces in the study area was CNY 3.28667 trillion in the same year, accounting for 28.73% of the national GDP. In this area, there are four distinct seasons: spring (March through May), summer (June through August), autumn (September through November), and winter (December, January, and February).

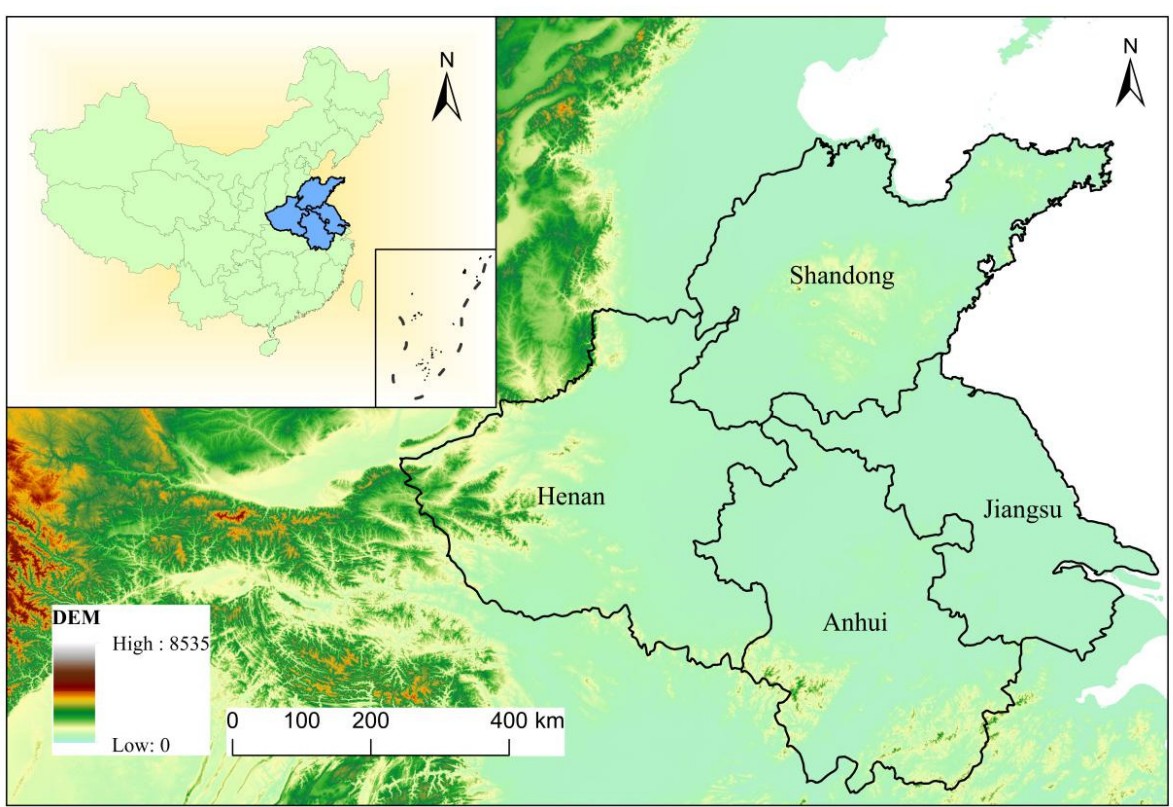

**Figure 1.** Geographical and topographical location map.

## 3. Data and Methods

### 3.1. Data Source and Processing

#### 3.1.1. Satellite Data

This study used data from the ozone monitor (Ozone Monitoring Instrument-OMI) on the Aura satellite. The OMI data is divided into four levels. The OMI sensor's secondary data products are used in this study, products include: $O_3$, cloud, aerosol and surface ultraviolet radiation products. OMI is a new-generation atmospheric composition detection sensor after GOME and SCIA2MACHY. It has a scanning width of 2600 km, a spatial resolution of 14 km $\times$ 24 km, and a wavelength range of 270–500 nm, the data of $NO_2$, $SO_2$ and HCHO used in this paper all come from this sensor.

#### 3.1.2. Aerosol Data

A NASA Goddard Earth Science Data and Information Service GES-DISC aerosol product with a spatial resolution of 13 km $\times$ 24 km and a temporal resolution of 1 day From January 2010 to December 2020, SSA, AOD, and AAOD at 388 nm, as well as SSA and aerosol species at 500 nm, were extracted. OMI offers three types of aerosols: smoke aerosols, dust aerosols, and sulfate aerosols. When there are multiple aerosol types in the extraction range, the region's aerosol type is defined as a mixed aerosol type. The mixed aerosol data are used for correlation analysis in this paper.

### 3.1.3. DEM Data

The DEM data used in this paper are derived from the global digital elevation model (GDEM) version 2.0. The DEM data set is a high-precision global elevation data set based on NASA Earth Observation Satellite Terra observations. Global elevation data are clipped according to the contour vector map of the study area using ArcGIS, and a DEM image is obtained.

### 3.2. Research Method

### 3.2.1. Coefficient of Variation

It is used to quantify the degree of stability of data to a certain extent and to reflect the degree of relative discrete fluctuation of data. We may therefore examine the stability of the spatiotemporal variation of pollutant column concentration by computing the coefficient of variation of each pixel's column concentration. The formula for its computation ((1) and (2)).

$$\sigma = \frac{1}{x}\sqrt{\frac{\sum_i^n (x_i - \bar{x})^2}{(n-1)}} \tag{1}$$

$$Cv = \frac{\sigma}{\bar{x}} \tag{2}$$

In the formula, $Cv$ is the coefficient of variation, $x$ is the average value, and $\sigma$ is the standard deviation. The smaller the value of $Cv$, the less fluctuation there was in the time series data and the better its stability; otherwise, there was a great fluctuation in data.

### 3.2.2. Interpolation Analysis

In this article, the Kriging interpolation method is employed in the process of processing pollution data because the average calculation approach cannot handle long time-scale data and the results may contain certain inaccuracies. Priority is given to the spatial variation distribution of spatial attributes, and the nearby data is estimated using the data from several known points. Using the variation function as a tool, the weight value of the known points' parameters around the estimated points is then established, and the best and most accurate estimation of the estimated points is made. The examination of a region as a whole benefits from the use of the known data to estimate a range of other variables. The formula for its computation ((3) and (4)):

$$\hat{Z} = \sum_{i=1}^{n} \lambda_i z_i \tag{3}$$

where $\hat{Z}$ is the estimate at point $(X, Y)$, that is, $Z = Z(X, Y)$. $\lambda_i$ is the weight coefficient, which is also used to estimate the value of the unknown point by the weighted sum of the data of all the known points in space. However, the weight coefficient is not the reciprocal of the distance, but a set of optimal coefficients that satisfy the minimum difference between the estimated value $\hat{Z}$ at the point $(X, Y)$ and the true value $Z$, namely:

$$\min_{\lambda_i} Var(\hat{Z} - Z) \tag{4}$$

### 3.2.3. Backward Trajectories and Cluster Analysis

The Joseph-Louis Lagrange HYbrid Single-Particle Lagrangian Integrated Trajectory model (HYSPLIT) is a system for computing and analyzing the air movement, deposition, transport, and diffusion trajectories of atmospheric pollutants [21,22]. It was created collaboratively by NOAA and the Bureau of Meteorology (BOM), and for more than 30 years, it has been used to compute and analyze the transport and diffusion paths of dangerous compounds. Applications include modeling of their propagation, diffusion, and deposition [22,23] as well as backward trajectory analysis of pollutants and dangerous

compounds [24–26]. One of the most often used atmospheric transport and diffusion models in atmospheric science is still HYSPLIT.

### 3.2.4. FNR Indicator

As a measure of $O_3$ sensitivity, the FNR (HCHO/NO$_2$) indicator has some degree of scientific validity [27,28]. In their study of the $O_3$ sensitivity of various US cities, Duncan et al. [29] used this method to analyze the higher-resolution OMI (Ozone Monitoring Instrument) satellite data. They discovered that in the Los Angeles area, when the HCHO/NO$_X$ ratio was less than 1, the $O_3$ sensitivity of the area, when the ratio of HCHO/NO$_X$ was between 1 and 2, can be considered VOCs-NO$_X$ cooperative control, and when the ratio of HCHO/NO$_X$ was.

### 3.2.5. The Reliability Test and Mann–Kendall Trend Correlation Coefficient

The Mann–Kendall trend test is a non-parametric statistical test method recommended by the World Meteorological Organization (WMO) and has been widely used. Its advantage is that the sample does not need to obey a certain distribution and is not disturbed by a few outliers. WMO has a high degree of quantification and a wide detection range, so it is more suitable for the examination of the sequence variables and type variables.

In this paper, the spatial correlation of air pollution is studied using a statistic called the Kendall grade correlation coefficient ($\tau$), is a statistic used to measure the association between two measured quantities. Positive correlation, negative correlation, and uncorrelation are the three categories into which the results can be separated. The reliability analysis shows that approach and 0 has greater reliability than approach and 1, which has lower dependability. The formula for its computation (5):

$$\tau = \frac{N_c - N_d}{n * (n-1)/2} \tag{5}$$

where $N_c$ represents the number of consistent, $N_d$ represents the number of inconsistent, $n$ represents the number of elements.

## 4. Results and Discussion

### 4.1. Multi-Pollutant Time Change

Figure 2 is a line graph of monthly changes using the remote sensing data of five pollutants from January 2011 to November 2020. The blue solid line is the overall change trend line of the pollutant, the equation is the trend line equation, the blue broken line is the average value, the red broken line is the maximum value, and the green broken line is the minimum value. NO$_2$ has a relatively obvious change cycle. It reaches the highest value from December to January of the following year, then starts to decline, and reaches the lowest value in July each year. Some months will appear higher than the previous month or the next month. The overall change shows a downward trend year by year. SO$_2$ had no discernible change cycle before January 2016, and the maximum fluctuation range was large. After 2016, the change cycle became stable and the maximum value and the gap between the minimum value and the average value also began to decline around January each year. The highest value is achieved and the lowest value is reached in July. Absorptive aerosols (UVAI) had an understandable change cycle, reaching the highest value in January each year and reaching the lowest value in July each year, and the overall change trend was not obvious. The change cycle of $O_3$ is stable, reaching the highest value between March and April every year, and reaching the lowest value between November and December every year, appearing in a slight downward trend as a whole. The change cycle of HCHO is likewise relatively stable. Synthesizing the temporal change process of five different pollutants, the different high and low values in different periods are mainly related to local temperature, illumination, energy consumption, etc. From July to March of the following year, the temperature gradually decreases. The fuel required for heating has greatly increased, resulting in intensified emissions of NO$_2$, SO$_2$ and UVAI in the region.

From March to August, the temperature gradually increased, the solar radiation increased, the rainfall increased, the vegetation coverage increased, and HCHO and $O_3$ emissions intensified.

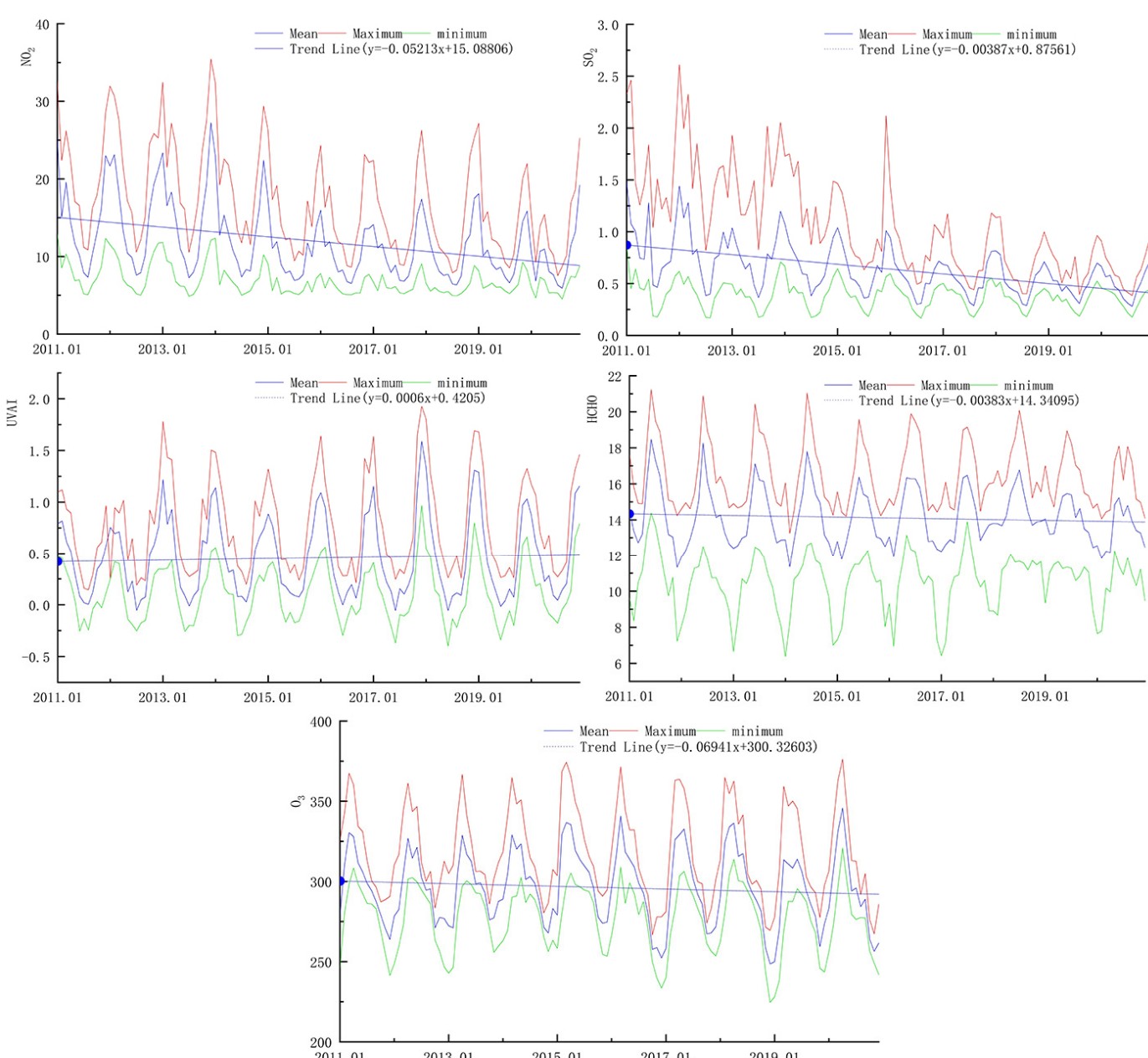

**Figure 2.** Monthly maximum, minimum and mean values of five air pollutants.

### 4.2. Spatial Distribution of Multi Pollutants

Figure 3 displays the geographic distribution of the four pollutants in the study area. When the annual average distribution of HCHO changed, the high-value areas were primarily concentrated in the northern Shandong province and the border area between Shandong and Henan province. These two high-value areas served as the center of the distribution, which gradually decreased to the surrounding areas. Eastern Henan, northern Anhui, and a tiny portion of western Jiangsu were all included in the sub-high value area. The research area's HCHO concentration was at its lowest during the winter, and it was at its highest during the summer (June, July, and August). The distribution law of North High and South Low is shown.

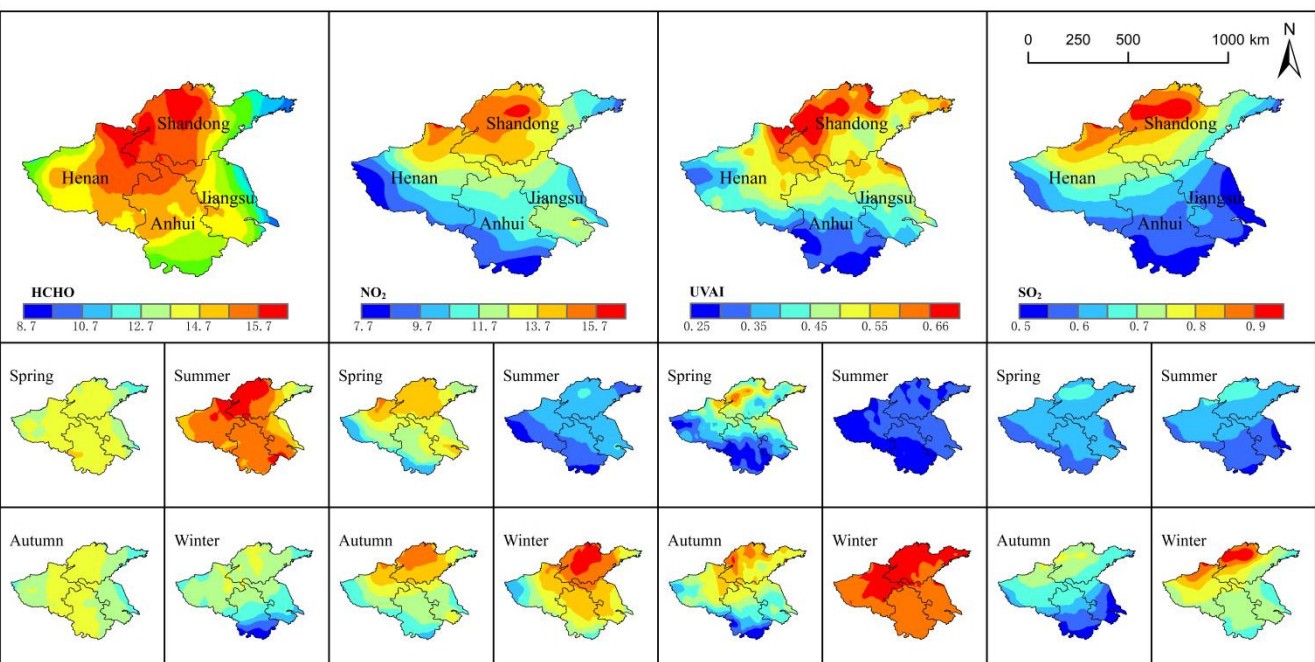

**Figure 3.** Spatial distribution of total HCHO, NO$_2$, SO$_2$ and UVAI and seasonal spatial distribution of column concentration.

The study area's northern Shandong province was home to the majority of the high-value areas in the fluctuation of the yearly average distribution of NO$_2$. Overall, there was a tendency for values to be higher in the north and lower in the south; the low-value area is primarily found in the southwest of the provinces of Anhui and Henan, with values rising from the southwest to the northeast. Winter (December, January, and February) north of the study area had the highest concentrations of NO$_2$, whereas summer and the southwest of the research area had the lowest concentrations.

The low-value area was in the south and gradually shrank from north to south in the mean annual distribution change of UVAI. The high-value area was concentrated in the north. We can see that UVAI is more seasonal and extreme than the other three air pollutants, with the lowest values in summer covering the entire study area and the highest values in winter covering the entire study area. Autumn and spring show the main trend of high north and low south. Liaocheng was the main area, and the west of Shandong province was the main high-value area.

The annual distribution of SO$_2$ was more symmetrical, with Liaocheng having the highest value and Anhui and Liaocheng provinces having the lowest value. Winter had the highest SO$_2$ column concentration among the four seasons. Winter was followed by autumn (September, October, and November), while spring and summer were comparable. In all four seasons, the north had a high SO$_2$ column concentration whereas the south had a low one.

### 4.3. A Comparison of the Evolution of Various Contaminants before and after COVID-19's Onset

After the outbreak, many countries and regions have adopted a series of policies and regulations to limit the spread of the virus, including implementing strict quarantines, prohibiting large-scale private and public gatherings, restricting private and public transportation, encouraging social distancing, implementing curfews, and even block the entire city. Locking down cities can greatly improve environmental quality, and the more industrialized cities have more significant air quality improvements during the blockade [30].

First, assess the impact of the epidemic on the emission of air pollutants. Taking Liaocheng, a key air pollution city in the study area, as an example, the Liaocheng atmospheric AQI (Air Quality Index) index is used to analyze the increase or decrease rate of

AQI before and after the epidemic. Make a distinction chart of the monthly change rate of the Liaocheng AQI index from January 2017 to December 2021(Figure 4). The maximum decline rate occurred in March 2020, reaching −55.43%, which was the month with the largest decline since 2017. This decline far exceeded the decline from February to March in previous years. It originated from the shutdown of local public enterprises and production during the epidemic, and the residents managed at home. It can be shown that the epidemic has indeed brought a positive impact on regional air quality. Therefore, we continue to analyze the overall changes of various pollutants in the study area in 2020 compared to 2019.

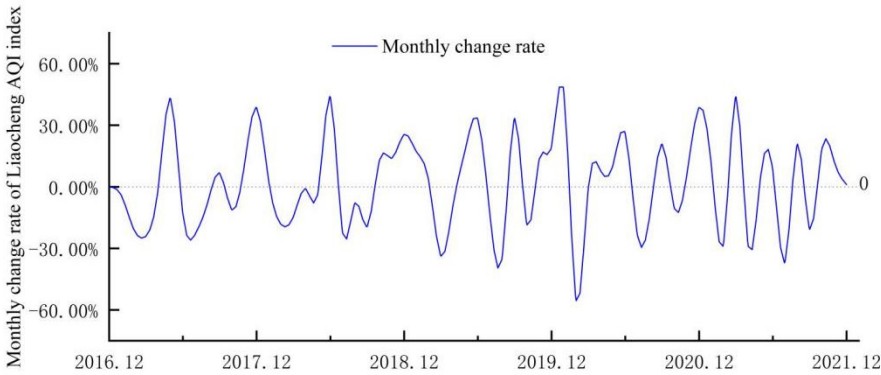

**Figure 4.** Monthly change rate of Liaocheng AQI (Air Quality Index) index from January 2017 to December 2021.

The different grades in the epidemic scenario before and after the change were calculated by comparing changes in four air pollutants between the year before and the year after the breakout of COVID-19, as well as an examination of how much human activity affects air pollution. As can be seen from Figures 5 and 6, the four pollutants have greatly reduced, four pollutants in high-value areas concentrated in the north of the research area will now be discussed one by one, along with the findings of the calculation. The seventh level area in 2020 was 45% smaller than it was in 2019, the eighth level area was 5% smaller than it was in 2019, and the eighth level area was 17% smaller than it was in 2019. The eighth-level area was 17% smaller in 2020 than it was in 2019, and the seventh-level area was also 17% smaller. The eighth-level area was 4% smaller in 2020 than it was in 2019, but the seventh-level area was 13% larger. In 2020, the UVAI eighth-level area was 8% smaller than it was in 2019, and the seventh-level area was 14% smaller (Figures 5 and 6).

In conclusion, the areas of HCHO, $NO_2$, and UVAI dropped in 2020 compared to the areas of the 2019 high-value region, with HCHO experiencing the greatest decrease, followed by $NO_2$. The size of the 2020 high-value region declined dramatically when compared to that of the 2019 high-value region, despite a decrease in the $SO_2$ eighth level, while the seventh level grew. To slow the spread of the virus after the breakout in China, the Chinese authorities put an embargo on the entire nation. All forms of transportation by land, ocean, and air have come to a complete halt. No discernible decrease in $SO_2$ occurred during the process. The area of high-value areas of numerous air pollutants decreased in 2020 compared to 2019 primarily because the main $SO_2$ emissions were released during the regular operation of heating in the winter while other sources of air pollution stopped emitting. It is undeniable that COVID-19 has had a detrimental effect on both human health and the global economy, but it has also helped to reduce air pollution.

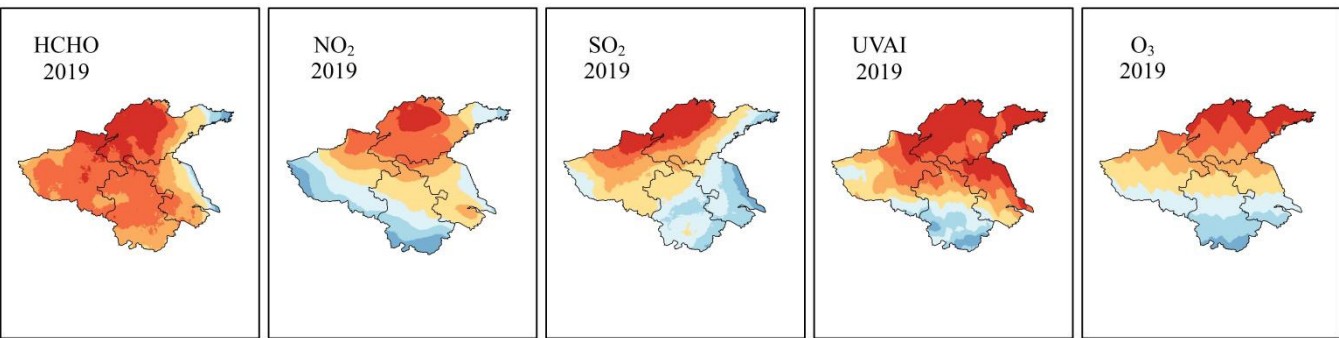

Concentration distribution of four kinds of air pollutants (In 2019 the COVID-19 has not outbreak )

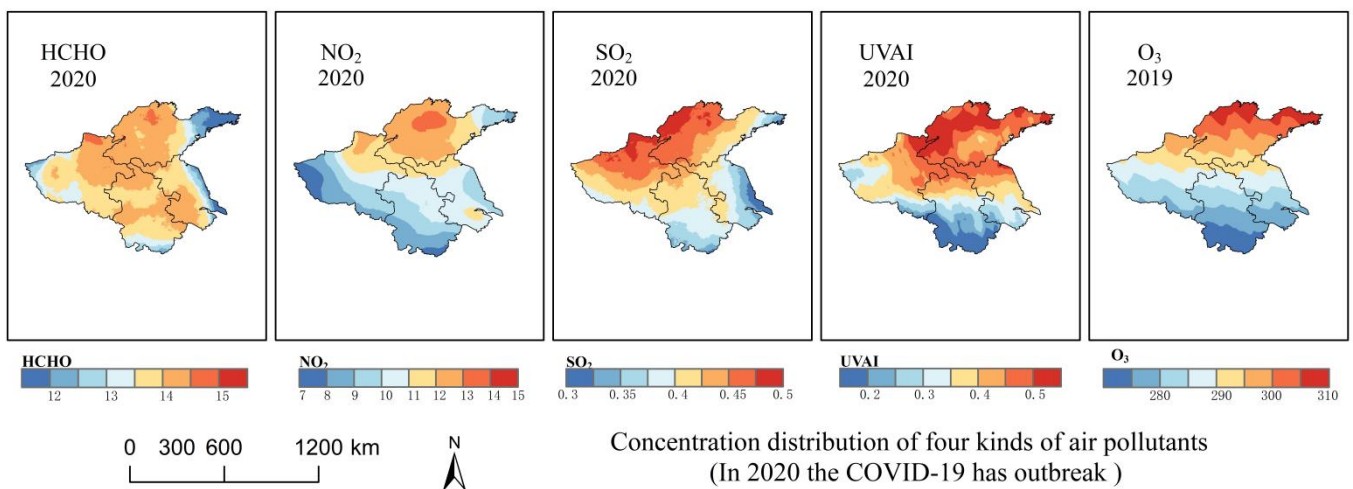

Concentration distribution of four kinds of air pollutants
(In 2020 the COVID-19 has outbreak )

**Figure 5.** Comparison of spatial distribution and change of five kinds of atmospheric pollution column concentrations before COVID-19 in 2019 and after COVID-19 in 2020.

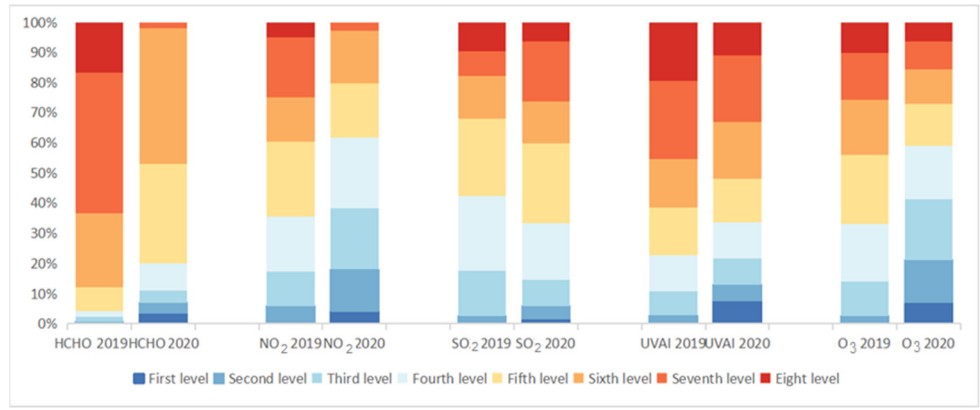

**Figure 6.** Percentage change of five air pollutants in different grades.

### 4.4. Spatial Stability of Multi Pollutants

By determining the coefficient of variation for four different contaminants in the research area, the spatial stability of several pollutants was examined (Figure 7). There were no obvious differences in the stability of HCHO, and the CV coefficient for the entire region was less than 0.05, indicating that HCHO tended to be stable in terms of year-over-year change. The stability of $NO_2$ ranged from 0.2 to 0.1, and the stability was generally stable; the CV coefficient for the vast majority of the study area is approximately 0.15. The small-area area in the west of Henan Province has a wider range of fluctuation than the rest of the region, while the stability of $SO_2$ exhibits obvious strip change. The stability of UVAI

also does not exhibit obvious fluctuation, and the relative fluctuation amplitude in the large area is small. The CV coefficient gradually decreased from north to south, with less area in southern Henan, higher stability, and smaller fluctuation in southern Anhui, and southeastern Jiangsu. The CV coefficient of the small-area area in the north of Shandong province and the north of Henan province is above 0.3, the fluctuation is greater and the stability is poor. $O_3$ is extremely stable throughout the entire region, with little volatility and stability lower than 0.05. The spatial-temporal distribution pattern of atmospheric multi-pollutants in Jiangsu Shandong Henan Anhui urban agglomeration in 2010–2021 is composed of the spatial-temporal distribution of the remaining four pollutants in the study area, with the exception of $SO_2$, which is unstable in some locations.

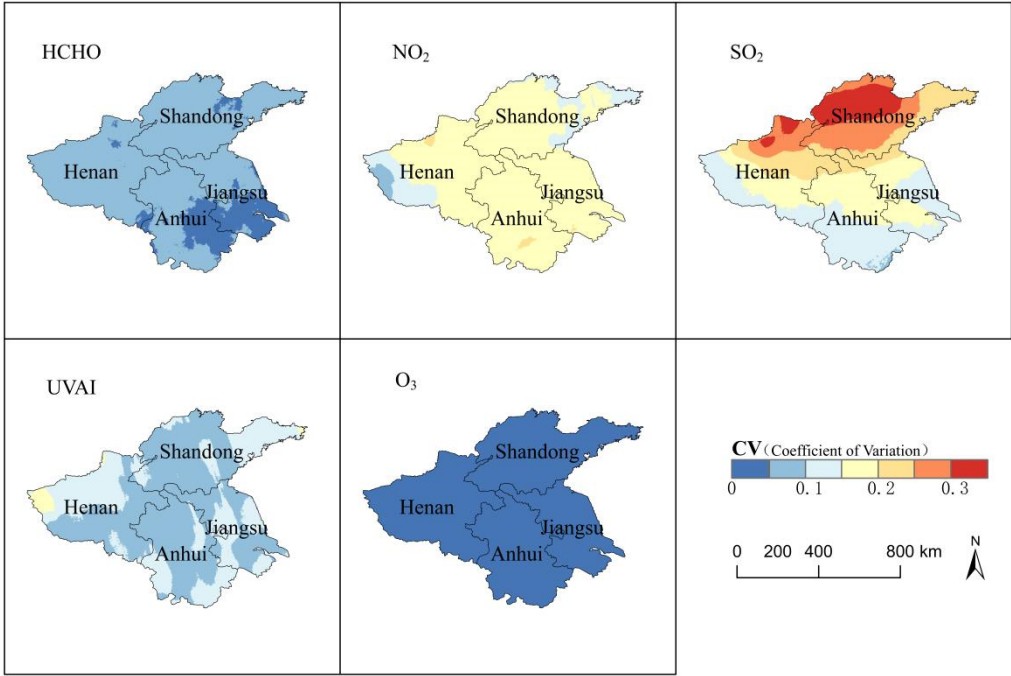

**Figure 7.** Spatial distribution of variation coefficients of 5 air pollutants.

### 4.5. Transmission Path Analysis of Multi Pollutants

It was established from the aforementioned analysis that the high-value point in the northern part of the study area was quite close to that in the northern part of the study region. According to the data released by the Shandong Provincial Department of Ecology and Environment in April 2021, there are 168 cities in the country, of which Heze, Zibo, Linyi, and Liaocheng in Shandong province are in the bottom position in the country; combined with the four-pollutant distribution maps, it was determined that the high-value areas of various pollutants were in Liaocheng City; therefore, they were set as high-value points. The backward trajectory model was used to simulate the atmospheric transport in Liaocheng throughout the four seasons (Figure 8). The air mass trajectories in various seasons were examined using cluster analysis after a significant number of transport trajectories were obtained. There were five key directions in which the cluster analysis was performed. When compared to the northwestern direction, the southwestern direction of air mass transport was shorter, accounting for 26.97% of the total. In spring, the air mass from the northwest to Inner Mongolia and Mongolia was more abundant, with the green and red trucks accounting for 35.96%, the yellow tracks from the north 19.1%, and the southeast 17.98%. Northeast air mass transport accounted for 5.43% of summer air travel, while the northwest air mass transport reduced to 17.39%. It was observed that more transportation came from the southeast coast in summer, including long- and short-distance travel with the predominating coastal sea–land breeze. However, the air mass transport

from the southeast ocean area increased, accounting for 48.1% in both offshore and open seas. In the fall, 42.58% of the tracks originated in the south, while 57.42% were from the northwest. Autumn has a dominant northwest breeze, although the south has no significant airflow. Winter air masses were still dominated by those from the northwest, making up 59.89% of the total, while those from other directions were weaker. In conclusion, the majority of the year's long-distance air traffic originated from the northwest of Liaocheng, whereas the majority of the province's short-distance air traffic originated from its local region and the southeast coast. Inner Mongolia, in the northwest, is an area with abundant and high mineral deposits from which it is simple to transport significant quantities of sand to the Shandong Peninsula. Air currents transport weighty volumes of pollutants into Shandong province from the Beijing–Tianjin–Hebei region where the industry is well developed.

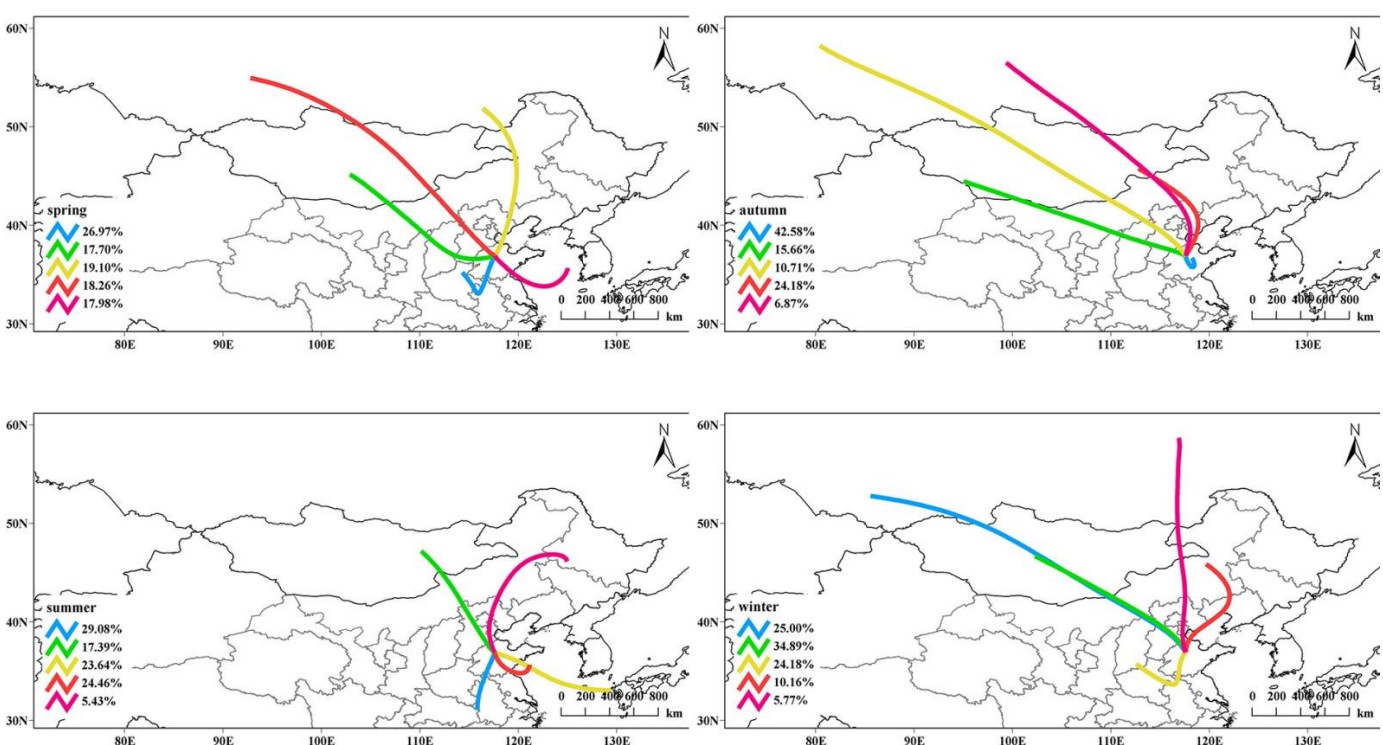

**Figure 8.** Four seasonal backward trajectories and their proportions of high-value points in the study area.

## 4.6. Multi-Pollutant Correlation Analysis

According to the data obtained from the remote-sensing monitoring of $O_3$, the months of March to September were split into $O_3$-occurrence periods, while the remaining months were non-$O_3$-occurrence periods. This study calculated the spatial correlation between different air contaminants and $O_3$ during the $O_3$-generation period (Figure 9), and its reliability was verified. The results indicate that $O_3$ and HCHO have a strong positive link in the northeast and southwest of the research area, where it accounts for 18% of the total. On the other hand, the south and north have a strong negative correlation, where it accounts for 32.5% of the total with excellent reliability. The northeastern corner of the study area, where the positive correlation area accounts for 53% of the area, presents the highest reliability, and the south of the study area, where the negative correlation area accounts for 4% of the area, has the highest credibility value. However, the rest of the region presents neither a strong nor positive relationship with the region. The areas with a positive connection between $O_3$ and $SO_2$ are primarily in the eastern region of the research area. A total of 39% of the region presented a good correlation, and the dependability was

high. The mild, negative association between $O_3$ and UVAI was located to the south of the study area, while a positive correlation was focused in the study area's northeastern corner and center, which accounted for 45% of the entire region. The method by which various contaminants participate in the production of $O_3$ is primarily responsible for the correlation discrepancy. This process is examined further in the subsequent $O_3$-sensitivity investigations performed in this study.

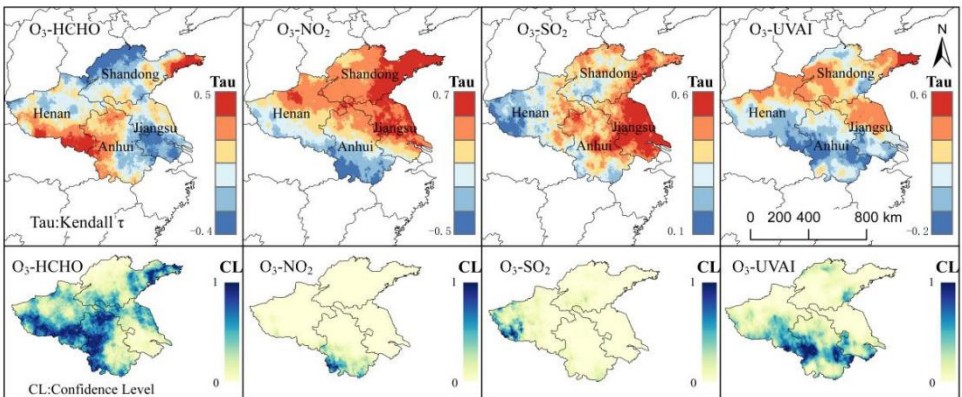

**Figure 9.** Spatial correlation and reliability test of four pollutants and $O_3$ during $O_3$ occurrence period.

*4.7. $O_3$ Sensitivity Analysis*

The main sources of $O_3$ production in the study area are identified using FNR indicators based on satellite data products. Seasons are distinguished primarily to highlight differences between them, and the distribution and content of possible $O_3$-producing variables vary according to the season. When the FNR ratio is less than 1, VOCs control can be assumed, when the ratio is between 1 and 2, VOCs-$NO_X$ cooperative control can be assumed, and when the ratio is larger than 2, $NO_X$ control can be assumed.

The $O_3$ control area was calculated using the 10-year mean value of the four seasons to obtain the main long-term control pollutants in different areas in the study area. The seasonal FRN values of the study area (Figure 10) reveal that in the spring, Shandong, northern Henan, and southeastern Jiangsu are VOCs-controlled areas, accounting for 18% of the total area, while southwestern Henan and southern Anhui are $NO_X$-controlled areas, the VOCs-$NO_X$ coordination-control area accounted for 3%, and the VOCs-$NO_X$ coordination-control area only accounted for 26% in. In the fall, VOCs were only in control of a tiny portion of Shandong province; VOCs-$NO_X$ was in charge of the remainder. In the winter, VOCs were under cooperative control with $NO_X$ in the southwest of Jiangsu province and the southeast of Anhui province, making up 44% of the overall area.

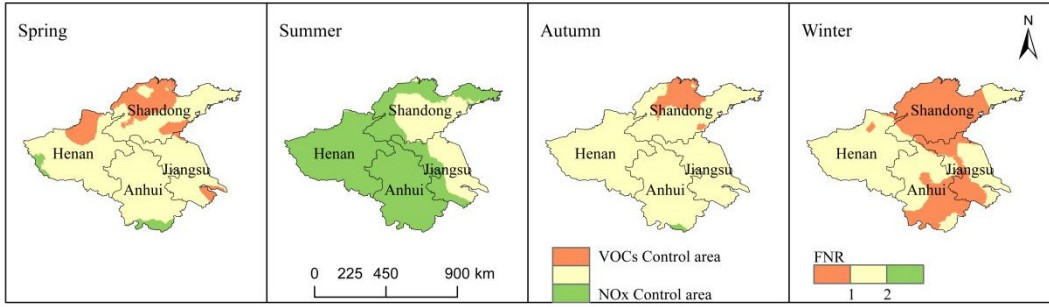

**Figure 10.** $O_3$ sensitivity zoning in four seasons based on FNR index.

In conclusion, because the amount of HCHO in the air is lower in the spring, fall, and winter than it is in the summer, the study area is primarily a VOCs control area and a cooperative control area. In the VOCs Control Area, where $NO_X$ is more prevalent,

the rise in VOCs creates the conditions for the production of $O_3$, making VOCs the main controlling factor, it is crucial to control organic emissions in the area. Since there is a high concentration of HCHO in the air during the summer, the study region is primarily in an area under $NO_X$ management. Since HCHO is abundant in the $NO_X$-controlled area, it is necessary to strengthen $NO_X$ control to prevent the creation of $O_3$.

## 5. Conclusions

From the present study, the main conclusions are as follows: First, from 2011 to 2021, air pollutants decreased, with $SO_2$ levels exhibiting the greatest and UVAI levels the least decline; the high values of several pollutants were mainly concentrated in the northern part of the study area. Second, the main source of pollutants in the high-value point was the long-distance transmission in the northwest region in spring and winter, and the main source was the short-distance transmission in the south and southeastern coastal regions in summer and autumn. Thirdly, through the study on the spatial correlation of air pollutants during the production of $O_3$, it was observed that $O_3$ and HCHO were mainly negatively correlated. The correlation between $O_3$ and $NO_2$ was positive in the northeastern direction and negative in the southern direction. $O_3$ and $SO_2$ were positively correlated. The correlation between $O_3$ and UVAI was small. Fourthly, the FNR index method was used to divide the $O_3$-controlled areas in the study area. It was observed that the northern part of Shandong province was a volatile organic compound-controlled area in spring and autumn. In the summer, a large area west of the study area was controlled by $NO_2$. Finally, the concentrations of HCHO, $NO_2$, and UVAI were compared before and after the outbreak. Based on the above conclusions, the concentration changes, time and space changes, transport paths, and the relationship with $O_3$ of these five air pollutants in the study area can be better understood; however, a considerable amount of research is still necessary to understand and solve the local atmospheric environmental problems that exist. At the same time, we also know that the city blockade can considerably reduce the level of air pollution, but the high economic cost of this is not beneficial to solve the pollution problem; there are many other cheaper ways to achieve the same environmental goals.

**Author Contributions:** Conceptualization, T.J. and B.P.; methodology, T.J.; software, B.P.; validation, T.J., B.L. and B.P.; formal analysis, J.W.; investigation, B.P.; resources, T.J.; data curation, S.L.; writing—original draft preparation, B.P.; writing—review and editing, B.P.; visualization, M.L. and S.P.; supervision, T.J.; project administration, T.J.; funding acquisition, T.J. All authors have read and agreed to the published version of the manuscript.

**Funding:** This work was supported by the National Natural Science Foundation of China (2016YFC0500907) and the Natural Science Foundation of Gansu Province (CN)(17YF1FA120) at the Key Laboratory of Resource Environment and Sustainable Development of OasisGansu Province.

**Institutional Review Board Statement:** Not applicable.

**Informed Consent Statement:** Not applicable.

**Data Availability Statement:** All the data generated or analyzed during this study are included in this published article.

**Acknowledgments:** Throughout the writing of this dissertation I have received a great deal of support and assistance. I would first like to thank my tutors, Tianzhen Ju, whose expertise was invaluable in formulating the research questions and methodology. Your insightful feedback pushed me to sharpen my thinking and brought my work to a higher level. I would particularly like to acknowledge my team members, Bingnan Li, Jiwei Wang, Shuya Liu, Meng Li, and Shuai Peng, for their wonderful collaboration and patient support.In addition, I would like to thank my parents for their wise counsel and sympathetic ear. You are always there for me.

**Conflicts of Interest:** The authors declare that they have no known competing financial interests or personal relationships that could have appeared to influence the work reported in this paper.

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
