# Peer review of "A Characteristic Analysis of Various Air Pollutants and Their Correlation with O3 in the Jiangsu, Shandong, Henan, and Anhui Provinces of China"

_sustainability, doi:10.3390/su142113737_

Round 1

Reviewer 1 Report

The article fits the purpose of the journal; however, there are several issues to be fixed:

a) The introduction is way to focused. You would need to amplify your discussion in order to attract more readers. Discuss on the use of low-cost sensors for measuring outdoor pollutants. Some references.

Rai, A.C., Kumar, P., Pilla, F., Skouloudis, A.N., Di Sabatino, S., Ratti, C., Yasar, A. and Rickerby, D., 2017. End-user perspective of low-cost sensors for outdoor air pollution monitoring. Science of The Total Environment607, pp.691-705.

Baldelli, A., 2021. Evaluation of a low-cost multi-channel monitor for indoor air quality through a novel, low-cost, and reproducible platform. Measurement: Sensors17, p.100059.

Kumar, P., Morawska, L., Martani, C., Biskos, G., Neophytou, M., Di Sabatino, S., Bell, M., Norford, L. and Britter, R., 2015. The rise of low-cost sensing for managing air pollution in cities. Environment international75, pp.199-205.

Besides formaldehyde is a very interesting. It would be nice the comparison with other studies:

Alonso, M.J., Madsen, H., Liu, P., Jørgensen, R.B., Jørgensen, T.B., Christiansen, E.J., Myrvang, O.A., Bastien, D. and Mathisen, H.M., 2022. Evaluation of low-cost formaldehyde sensors calibration. Building and Environment222, p.109380.

Baldelli, A., Jeronimo, M., Tinney, M. and Bartlett, K., 2020. Real-time measurements of formaldehyde emissions in a gross anatomy laboratory. SN Applied Sciences2(4), pp.1-13.

and many more.

b) Figure 2 and 4 needs to be improved in quality

c) You need to summarize more over the text, there are a lot of repetitions and, sometimes, the poor use of English grammar.

Reviewer 2 Report

The paper has following problems. It is necessary to solve them one by one in your revised paper.

(1)    Introduction needs to be further added and improved. The current status of air pollutants and O3 should be presented from different perspectives. The relevant literature on their research should be added.

(2)    The innovation point is not briefly explained in the paper. Please propose motivation and innovation for studying the relationship between air pollutants and O3.

(3)    The experimental method needs to be further introduced with relevant equations, such as the Kriging interpolation method. The research methods used in the article need further explanation.

(4)    Please adjust the format of the article carefully, such as the format of references and images.

(5)    The conclusion section contains too much content and does not summarize and focus.

Round 2

Reviewer 2 Report

The paper is revised and is better than the first version. However, it can be further improved. For instance, the contribution of the paper should be presented one by one in the Introduction Section.

The organization of the paper is not given in the Introduction Section.
